# The Possible Role of Postnatal Biphasic Dysregulation of IGF-1 Tone in the Etiology of Idiopathic Autism Spectrum Disorder

**DOI:** 10.3390/ijms26104483

**Published:** 2025-05-08

**Authors:** András Visegrády

**Affiliations:** Pharmacology and Drug Safety Research, Gedeon Richter Plc., Gyömrői út 19-21, H-1103 Budapest, Hungary; visegrady@gedeonrichter.com; Tel.: +36-1-889-8741

**Keywords:** autism, brain overgrowth, insulin-like growth factor 1, intrauterine growth restriction, insulin resistance

## Abstract

Autism spectrum disorder (ASD) is a pervasive condition of neurodevelopmental origin with an increasing burden on society. Idiopathic ASD is notorious for its heterogeneous behavioral manifestations, and despite substantial efforts, its etiopathology is still unclear. An increasing amount of data points to the causative role of critical developmental alterations in the first year of life, although the contribution of fetal, environmental, and genetic factors cannot be clearly distinguished. This review attempts to propose a narrative starting from neuropathological findings in ASD, involving insulin-like growth factor 1 (IGF-1) as a key modulator and demonstrates how the most consistent gestational risk factors of ASD–maternal insulin resistance and fetal growth insufficiency–converge at the perinatal dysregulation of offspring anabolism in the critical period of early development. A unifying hypothesis is derived, stating that the co-occurrence of these gestational conditions leads to postnatal biphasic dysregulation of IGF-1 tone in the offspring, leading first to insulin-dependent accelerated development, then to subsequent arrest of growth and brain maturation in ASD as an etiologic process. This hypothesis is tested for its explanation of various widely reported risk factors and observations of idiopathic ASD, including early postnatal growth abnormalities, the pervasive spectrum of symptoms, familial predisposition, and male susceptibility. Finally, further directions of research are outlined.

## 1. Introduction

### 1.1. Background

Autism spectrum disorder (ASD) is a neurodevelopmental disorder characterized by deficits in social communication and the presence of restricted interests and repetitive behaviors [1,2,3]. ASD displays increasing prevalence trends [4,5] and poses a steadily increasing burden on society [6,7]. ASD is diagnosed based on behavioral symptoms, but its manifestations are notoriously heterogeneous [2,8]. Despite the high prevalence and increasing awareness, little is known about the etiology of ASD [9,10]. It is characterized by a clearly demonstrated familial risk and high heritability [11,12], and numerous genetically defined syndromes display high comorbidity with autism. Large-scale genetic studies searched for genetic background in ASD, and a high number of candidate genes have been identified with high confidence, including among others *CHD8*, *TSC2*, *NRXN1*, *SHANK3*, *SHANK2*, and *MTHFR* [13,14,15]. Despite these advancements in genomic research, idiopathic ASD still represents the majority of cases [14,16]. The disorder is typically diagnosed in toddlers or in early childhood [17], although first manifestations seem to occur around 6–12 months of age [18,19]. The diagnosis of autism is rather stable throughout childhood [20,21] and adolescence [22,23], suggesting a critical role for early brain maldevelopment in the emergence of autism symptoms [24,25]. Therefore, in addition to the heterogeneity in symptoms, studies of ASD are also hampered by the timing of symptom onset, namely incidence during a period of intense brain development and maturation of the nervous system [26,27], requiring longitudinal observations and analysis of at-risk populations placed in a developmental framework [24,28].

Recognizing the proposed early origin of the disorder, environmental, familial, and gestational risk factors have been investigated extensively, and several risk factors of idiopathic ASD have been identified [1,29]. Among familial and environmental factors, an affected sibling [11,30] or male sex [31,32] are widely documented to substantially increase the risk for autism. Furthermore, increased maternal and paternal age have also been demonstrated as risk factors [33,34]. Among gestational and maternal conditions, the most robust and consistently identified risk factors for ASD are conditions related to maternal hypertension, namely preeclampsia and gestational or chronic hypertension [29,35,36]; maternal use of antidepressants, in particular, selective serotonin reuptake inhibitors (SSRIs) [29,37,38] and symptoms related to maternal metabolic disturbances like gestational diabetes [39,40,41,42] and pre-gestational overweight or obesity [29,43,44]. For perinatal factors, signs of fetal growth insufficiency, like preterm birth [40,45] or low or very low birth weight [40,46], are associated with ASD risk, but, interestingly, macrosomia (fetal overgrowth) also bears a slight risk for later ASD diagnosis [46].

Although ASD is typically not diagnosed before the age of 18 months, at the population level, several growth and developmental deviations have been described to precede or correlate with later ASD diagnosis. Accelerated early growth has been demonstrated for body length [47,48], total brain volume [25,49,50,51], cortical surface area [52], and excess extra-axial fluid volume [53]. Importantly, following a period of overgrowth in toddlers, a slowing or even arrested growth has been reported for many of these parameters in early childhood [51,52,54,55].

In addition to that observed in anthropometric measures, a biphasic developmental trajectory was also found for brain structural connectivity [56]. In particular, two highly valuable longitudinal imaging studies, comparing at-risk infants later diagnosed with ASD to non-affected peers [57] or diagnosed toddlers to typical controls [58], reported structural hyperconnectivity between brain regions at earlier scans, in contrast to hypoconnectivity at the later time points (24- ca. 36 months, depending on the study). In studies of head circumference, direct comparisons found early overgrowth in ASD (and in pervasive developmental disorder not otherwise specified, in earlier studies), which was absent in patients with attention deficit hyperactivity disorder (ADHD) [59,60] or generalized or other developmental delays [48,61].

Despite robust data on well-defined prenatal risk factors and postnatal developmental alterations with substantial comorbidity with ASD, to date, no theoretical framework exists for the etiology of this disruptive disorder that convincingly matches these observations. In recognition of the pervasive heterogeneity of the disorder, recently, multiple etiologies have been considered to account for the diverse manifestations of the dysfunctions [62]. However, it has yet to be demonstrated that symptoms group according to suggested pathomechanisms or risk factors [8]. Moreover, multiple disjunct etiologies or genetic causation have to account for the increasing prevalence of ASD, with the disorder retaining its heterogeneity of manifestations, stable pattern of risk factors, and characteristic developmental trajectory. Alternatively, instead of multiple etiologies or risk genes, a central pathomechanism could be considered in idiopathic ASD, a process that could inherently account for the heterogeneity of symptoms and be associated with known ASD risk factors. In the latter sections, such a hypothesis of idiopathic ASD etiology is proposed and discussed in light of scientific observations on ASD.

### 1.2. Neuropathological Findings in ASD and a Neurodevelopmental Narrative

Despite occasional and specific peripheral comorbidities of ASD, it is reasonable to assume that the brain is the most affected organ in the disorder. Neuropathology studies of autism patients report wide-ranging microscopic brain alterations [63,64,65,66] like decreased MECP2 expression [67] and widespread disorganization of cytoarchitecture and neural migration [68,69]. The most consistent microscopic observation, demonstrated in about 75% of cases, is the decreased number of Purkinje cells (PCs) in the cerebellum [63,65]–typically in the form of dispersed and partial loss [70,71]. The origin of this cell loss was believed to be prenatal, suggested by the lack of retrograde cell loss in the inferior olive [72], however, the presence of excess cerebellar basket and stellar cells [73] and the lack of hypoplastic folia [65] suggests a relatively late timing of this event, and therefore the precise time of atrophy could not be convincingly established yet (discussed in [65]). *Late postnatal* loss of PCs in autism has not been considered in the literature, due to the putative dependence of inferior olivary neuron survival on their PC targets [72]. However, two findings support the possibility of postnatal atrophy. First, the dependence of inferior olive neurons on their target PCs seems to weaken during postnatal development [74,75]. Namely, although retrograde inferior olivary numeric atrophy has been demonstrated in various mutant mouse strains with early PC loss, in strains with the latest postnatal onset of PC death (around postanal days 30 and 20 in *leaner* and *nervous* mice, respectively), little or no loss of inferior olivary neurons is reported [75,76]. Secondly, although retrograde atrophy of the inferior olive has been observed in human patients following acute cerebellar lesions [77], gradual and partial loss of PCs, in contrast, does not necessarily lead to detectable inferior olivary atrophy in adults [77,78]. Intriguingly, neuropathologic analysis of the youngest ASD specimens (3 to 4 years) revealed no PC loss [63,70,79], in line with the possibility of late postnatal atrophy. In summary, PC loss in ASD in later neonatal development cannot be excluded as a possibility and might even be in line with pathological findings [73]. If PCs were present throughout the regular development of the cerebellum, their loss might be associated with later changes in the internal environment, like the sudden loss of neurotrophic or survival factors, as PCs are known for their particular vulnerability to pathophysiological insults [80,81,82].

When searching for survival-promoting agents of PCs, insulin-like growth factor 1 (IGF-1) has been identified as a hormone having a pronounced and reproducible positive effect on PC survival [83,84] and development [85] both in vitro and in vivo [86,87]. The marked survival-promoting effect of IGF-1 on PCs seems specific in comparison to numerous other neurotrophic factors like nerve growth factor, brain-derived neurotrophic factor [88,89], ciliary neurotrophic factor [90], basic fibroblast growth factor, insulin, and insulin-like growth factor 2 [83], or to contradictory effects of neurotrophin-3 [88,89] or glial cell line-derived neurotrophic factor [86,91]. Thus, although other survival-promoting factors might also play a role, PC loss in ASD might be a result of late postnatal attenuation in the neurotrophic tone of IGF-1. Strikingly, in two highly valuable clinical datasets, liquor IGF-1 level of younger ASD patients, ages 1.9 years to 5 years, was found to be extremely low and lower than that of age-matched nonaffected patients [92] or neurotypical controls [93], a difference not found for nerve growth factor [94], insulin-like growth factor 2 or IGF-1 in ASD patients above the age of 5 [93].

Previously, loss of IGF-1 tone as an underlying mechanism in ASD has been suggested [95,96,97], and IGF-1 supplementation has been proposed for the treatment of ASD or related disorders [97,98,99].

### 1.3. Clinical Studies of IGF-1 and Related Factors in Neurodevelopmental Disorders

Multiple clinical trials have been performed investigating the effect of IGF-1 or related factors in neurodevelopmental conditions associated with ASD. Children of 5–9 years with Phelan-McDermid syndrome, all with *SHANK3* irregularities, were treated with recombinant human IGF-1 for 12 weeks in two consecutive studies [100]. Although treatment did not improve with statistical significance the primary endpoint, Aberrant Behavior Checklist—Social Withdrawal subscale in combined analysis, improvement was found in the Restricted Behavior subscale of Repetitive Behavior Scale—Revised scale and in sensory reactivity symptoms. Investigation of mecasermin (recombinant human insulin-like growth factor 1) in a double-blind study in Rett Syndrome in girls between 2 and 10 years for 20 weeks resulted in worsening of some of the primary endpoints with some improvement in measures of stereotypic behavior and social communication [101]. A pilot study with mecasermin in children with ASD (age 5–12 years) was terminated prior to completion.

In addition to IGF-1, analogues of the N-terminal tripeptide fragment (1–3) IGF-1, not a ligand of IGF-1R (IGF-1 receptor) but a factor influencing IGF-1 action, also entered clinical development. Trofinetide, a tripeptide analogue, was developed for Rett Syndrome with a significant improvement on the Rett Syndrome Behaviour Questionnaire [102] and was approved as a first-in-class drug for this indication in 2023. Additionally, it also showed a positive effect on Fragile X syndrome [103]. NNZ-2591, a synthetic analog of cyclic glycine-proline, the metabolite of (1–3) IGF-1, is currently in late-stage clinical development in multiple neurodevelopmental disorders.

The results of the above studies support a possible role of neurotrophic factors in the therapy of neurodevelopmental disorders, however, they also revealed several limitations. The timing and duration of the therapy in light of the potentially mitogenic effects of IGF-1 could be a concern despite lowered IGF-1 levels in patients [98]. Furthermore, the placebo effect in caregiver questionnaires might mask slight improvements in behavioral observations. Still, with efficacy in several genetic disorders using trophic factors, clinical development of further therapies might be accelerated.

### 1.4. IGF-1 and Its Dysregulation in Perinatal Complications

The hypothesis of IGF-1 deficit as the cause of ASD, unfortunately, does not account for the most consistent characteristics and risk factors of idiopathic ASD, such as early growth and connectivity anomalies, or various perinatal or familial risk factors. Furthermore, although IGF-1 regulates PC survival and development, IGF-1 knockout mice are not characterized by PC loss [104], which suggests a more complex dysregulation of IGF-1 in ASD.

IGF-1 is a trophic factor, a mediator of growth via its major molecular target, IGF-1R, ubiquitously expressed in most tissues, with the highest expression levels in neuronal cell types (Figure 1). Circulating IGF-1 is predominantly secreted from the liver but can be taken up by the brain, enabling modulation of central levels by circulating IGF-1 of liver origin [105,106]. IGF-1 is widely expressed also in other tissues [107] including the brain [84] and while paracrine action contributes to postnatal growth [108], liver-produced systemic IGF-1 also exerts substantial somatic growth-promoting effects [109,110], indicating that multiple sources of IGF-1 act jointly on tissue development and that endocrine IGF-1 also affects growth. Secretion of IGF-1 from the liver is predominantly regulated via the growth hormone-releasing hormone (GHRH)–growth hormone (GH)–IGF-1 axis [111], but peculiarly, perinatal regulation of endocrine IGF-1 is different, creating a vulnerability through a specific constellation of perinatal factors associated with later ASD risk. Finally, unlike the closely related insulin, IGF-1 is bound to a family of carrier proteins (insulin-like growth factor-binding proteins, IGFBP), which can regulate IGF-1 availability and thus exert further control on IGF-1 function in vivo.

In the fetus, IGF-1 release from the liver is regulated by circulating insulin with little effect on GH [113,114,115], also mirrored in the close-to-normal birth weight of GH-deficient infants [116]. In newborns, circulating IGF-1 is a determinant of somatic growth [117,118,119] and is associated with insulin level [120,121] with no or negative correlation to GH or GH binding protein [122,123], in line with typically no signs of growth deficit in congenital GH deficiency prior to 6–12 months [116]. Regulation of circulating IGF-1 in infants thus gradually changes from insulin-dependent to GH-dependent throughout the first year of life [123,124].

As IGF-1 is a major regulator of fetal growth, two groups of gestational maternal complications associated with metabolic and growth alterations in the offspring can lead to its dysregulation. Fetal growth restriction or intrauterine growth restriction (IUGR) often leads to a neonate with a birth weight lower than the 10th percentile. These small for gestational age (SGA) infants are characterized by short stature, decreased serum or cord IGF-1 level [125,126], and elevated insulin sensitivity at birth [126,127]. Their majority undergo a period of compensatory accelerated early postnatal growth (catch-up growth) [128,129] and reach a more insulin-resistant metabolic state [127,130,131]. An opposite alteration from the typical perinatal growth trajectory and metabolism is observed after macrosomia or fetal overgrowth. Macrosomic, or large for gestational age, newborns display hyperinsulinemia [132,133] and higher IGF-1 levels [134,135] in a more insulin-resistant state [133,136,137] and often suffer decelerated early postnatal growth–catch-down growth [138,139,140]. Catch-up and catch-down growth take place mainly within the first 6–9 months [138,139], the time window of insulin-dependent IGF-1 secretion in the newborn. SGA infants experiencing catch-up growth have elevated circulating IGF-1 levels compared to those remaining short or light [141,142].

These specific perinatal growth anomalies are highlighted because their maternal risk factors show a striking overlap with those of idiopathic ASD discussed in Section 1.1. In one group of conditions, SGA offspring and low birth weight are associated with maternal SSRI and antidepressant use [143,144], and IUGR is highly comorbid with preeclampsia [145,146] and gestational hypertension [147,148]. Conversely, in a second group of gestational conditions, macrosomia is strongly associated with maternal metabolic disturbances like gestational diabetes [149], obesity [150], or higher than recommended gestational weight gain [151], driven by insulin resistance [152]. Importantly, as discussed above, both groups of maternal conditions are strongly connected with ASD risk in the offspring [29,36,42,44].

## 2. Hypothesis

### 2.1. A Possible Etiological Role of IGF-1 Dysregulation in Idiopathic ASD

The gestational risk factors of ASD described above thus form two groups associated with opposing dysregulation of fetal growth, neonatal insulin sensitivity, circulating neonatal or cord IGF-1 level, and infant growth trajectory. More importantly, the timing of their postnatal consequences also overlaps with the earliest developmental alterations preceding ASD in infancy. Based on these overlaps, the novel hypothesis for idiopathic ASD presented here is that the simultaneous occurrence of these counteracting conditions during gestation could lead to a complex dysregulation of perinatal insulin homeostasis and insulin-mediated IGF-1 action in the affected offspring. Prenatally, these co-occurring factors might partially neutralize each other despite misalignment of fetal insulin sensitivity; however, after birth, isolated from maternal circulation, asynchronous compensatory processes could lead to biphasic dysregulation of neonatal insulin sensitivity and IGF-1 tone. In the first phase, separated from the placental unit, prevailing elevated insulin action in tissues could result in higher IGF-1 secretion, leading to accelerated growth in a sensitive period of brain development in infancy. Next, by the age of 12–15 months, maturation of the GHRH-GH-IGF-1 axis and sharp negative feedback due to elevated IGF-1 levels could lead to an abrupt drop in GHRH-GH-IGF-1 tone, and concomitantly to arrest of IGF-1-dependent developmental processes and slow maturation of the central nervous system (Table 1).

According to this hypothesis, thus, prenatal gestational anabolic disturbances result in a complexly dysregulated neonatal insulin-IGF-1 axis, leading first to elevated and later to decreased IGF-1 levels in affected infants, leading to somatic and brain overgrowth and subsequent growth arrest and delayed maturation (summarized in Figure 2). This biphasic postnatal growth differs from that of catch-up growth in SGA or low birth weight infants in that the latter is characterized by in utero growth restriction or insufficiency leading to delayed growth up to birth, while co-occurrence of maternal insulin resistance would partly compensate IUGR, resulting in a generally well-developed newborn with high capacity for overdevelopment in a postnatal compensatory growth spurt.

Since somatic growth anomalies, observed only in a subset of ASD patients, supposedly do not form the etiologic factors of ASD, core symptoms should depend on developmental alterations in affected brain regions. Brain development and connectivity of ASD brains seem to show peculiar dynamics at an early age [24,25,56]. Early connectivity abnormalities–mixed functional over- and underconnectivity–at 6 months were used to predict later ASD diagnosis with 100% specificity [153], and the alterations seem specific to ASD [154]. Additionally, brain overgrowth predominantly in cortical surface area by 6 and 12 months identified affected infants with high specificity (95%) [25]. As these developmental trajectories are distinct from those in global developmental delay [155] and ADHD [156] and as diagnosis stability of ASD is relatively high, it can be reasoned that this early period of brain overgrowth and probably overconnectivity is specific to autism and might form the pathomechanistic basis of idiopathic ASD [157]. The effect of overgrowth and subsequent arrest could be affected by genetic and environmental factors, and these, in concert, can contribute to the phenotypic heterogeneity of idiopathic ASD.

IGF-1R and its downstream PI3K/Akt/mTOR pathway have been shown to interact with processes associated with ASD and risk genes *TSC2* [158], *GIGYF1* [159], or *CNTNAP2* [160]. IGF-1 has been shown to elicit developmental effects in models of genes with strong ASD association, among others, *SHANK3* [161] and *DYRK1A* [162].

As described above, the potential role of IGF-1 in ASD has been recognized in the literature, and IGF-1 supplementation has been considered as a treatment for ASD [95,96,97,163,164]. The hypothesis of early postnatal elevated central IGF-1 tone or its biphasic dysregulation, however, has not been raised before. The above hypothesis thus represents a novel viewpoint of the role of IGF-1 in the appearance of idiopathic ASD.

### 2.2. Explanation of Observations and Risk Factors of ASD

IGF-1 is a pleiotropic modulator of growth [107] and brain development [165]; therefore, dysregulation of IGF-1 tone affects a broad range of neurodevelopmental processes and therefore could account for the most pervasive observation in ASD: the puzzling diversity of brain developmental, pathophysiological, and pathological findings, as well as the resulting variety of behavioral or neurological symptoms. According to this view, it is not a single site, but most of the organism and the central nervous system are affected by abnormal biphasic early overgrowth, and genetic and other factors could further shape the appearance of ASD. Since the highly arbored PCs of the cerebellum are known to display elevated vulnerability to pathophysiological conditions in vivo [80,82,166], their survival and development could be the most affected by altered central IGF-1 availability. Unprogrammed postnatal elevation of IGF-1 tone followed by an abrupt drop could lead to late and selective postnatal PC loss, without substantial atrophy of cerebellar basket cells, cerebellar stellar cells, and inferior olivary neurons. As cited above, the lack of PC loss in the youngest ASD cases subjected to neuropathological analysis is noteworthy, not contradicting the supposed PC loss throughout early childhood. This assumption is in line with the extremely low liquor levels of IGF-1 between 1.9 and 5 years in ASD patients [92,93]. Similarly, although peripheral and central IGF-1 do not necessarily correlate, lower serum levels of IGF-1 were found in ASD patients of 2–3 years of age compared to age-matched controls [167], and lower urinary secretion of IGF-1 has been reported in ASD in a very similar age range (2–5 years) [168]. Interestingly, in pediatric subjects above 5 years, cerebrospinal fluid (CSF) levels of IGF-1 did not differ between affected and control subjects [93], while serum levels were reported to vary in children with ASD, with various age groups typically ranging from 4 to 12 years [167,169,170,171].

The timing of the suggested biphasic alteration in circulating IGF-1 level coincides well with the ASD-specific switch in growth rate compared to controls [56,172] in light of the observations that neonatal somatic growth correlated with circulating IGF-1 levels [118,173]. In infants with later ASD diagnosis, excess extraaxial volume [53] and brain volume [52] seem to stabilize after 12–18 months. Similarly, although imaging studies reported structural hyperconnectivity prior to 20 months, hypoconnectivity was observed in ASD patients by the age of 2–3 years [57,58]. Additionally, a particularly interesting analysis has found that white matter overgrowth in ASD is specific to myelinization processes starting within the first 4 months after birth [174]. Based on the well-described effect of IGF-1 on brain development [165], the timing of these brain myelination periods overlaps well with the proposed timing of early postnatal developmental dysregulations in ASD.

The most consistent non-gestational risk factors of ASD could also be explained by the concept of perinatal IGF-1 dysregulation. Boys bear approximately a 3-fold risk of ASD incidence compared to girls [32]. Male infants are known to be born larger [175,176], have higher brain volume [177,178], and undergo slightly faster postnatal brain growth than females [26,179]. The IGF-1 level of newborn boys, in contrast, is reported as lower or similar to that of girls, both in serum [118,180,181,182] and CSF [183], and female infants are more insulin-resistant than boys [176,184]. Therefore, male infants, since they display a higher growth rate with lower or similar IGF-1 levels and concomitant higher insulin sensitivity than females, might be more sensitive to insulin-regulated hypertrophy mediated by excess IGF-1 and its abrupt drop, and this amplification of IGF-1-mediated effects might be the cause of their susceptibility to ASD.

The single highest risk for idiopathic ASD is an affected sibling [30] with a clear familial connection [12] despite the lack of high-penetrance single-risk gene variants [14]. Genetic architecture clearly modifies the penetrance of neurodevelopmental disturbances and thus definitely exerts a contribution to the appearance of ASD that should not be underestimated. Still, according to the hypothesis above, the combined incidence of two groups of gestational conditions could be the specific etiologic trigger for idiopathic ASD, and therefore, careful separation of genetic influence from the influence of the intrauterine environment is required. The contribution of genetic and environmental factors is most often assessed by analyzing concordance in monozygotic and dizygotic twins [12]. However, as intrauterine growth restriction, as presented above, can be suspected in the etiology of ASD, such a heritability analysis has to take into account the difference in placental organization of dizygotic and monozygotic twins. Dizygotic twins are obligatory dichorionic; they possess separate placentas, while the majority of monozygotic twin pairs are monochorionic, sharing the same placenta. Placental dysfunction is a major contributor to fetal growth insufficiency and IUGR [185,186]; therefore, monochorionicity might lead to overestimation of heritability as impaired placental function might not be separated from genetic overlap. It has to be noted that preeclampsia, the strongest maternal risk factor of ASD and a risk factor for IUGR, parallels idiopathic ASD in its familial aggregation with yet unclear genetic etiology [187].

The steeply increasing prevalence of ASD can be partly related to the increased awareness and diagnostic tolerance of the behavioral spectrum, however, a true increase in prevalence is nevertheless suggested [4,188]. According to the etiologic hypothesis presented, the co-occurrence of maternal insulin resistance and risk factors of IUGR, like chronic or gestational hypertonia, is a prerequisite for biphasic neonatal metabolic dysregulation. While prevalence trends in gestational hypertensive disorders in the last decades are reported as varied [189,190,191], the prevalence of pregestational obesity and pregestational/gestational diabetes has been rising significantly since decades [192,193,194,195] in line with the steadily increasing ASD prevalence.

Finally, increasing maternal age, too, is unambiguously a risk factor for metabolic syndrome–by multiple definitions–in women of childbearing age in developed countries [196,197], and also for its various manifestations like gestational diabetes [192,198], overweight, or obesity [199]. Increasing maternal age, especially above 40 years [34], as a risk factor for ASD could therefore indirectly transmit the effect of these gestational conditions.

### 2.3. Prospective Testing of the Hypothesis

The etiologic hypothesis presented here attempts to add points for consideration in order to improve our understanding of this pervasive disorder. Numerous epidemiologic studies have investigated the contribution of environmental, gestational, and maternal conditions to ASD risk. The principle proposed above is that of a combination of gestational conditions leading to opposing postnatal growth dysregulation (e.g., maternal metabolic disturbances in combination with preeclampsia, gestational hypertension, or SSRI use). Therefore, investigation of combinations of these specific conditions in regard to ASD incidence risk would be interesting, as suggested in general in [35]. Additionally, as signs of maternal metabolic syndrome and insulin resistance are risk factors of ASD [42] and as gestational diabetes bears a significant risk of later type 2 diabetes of the mother, the apparent risk posed by having an ASD offspring on later maternal type 2 diabetes is worth investigating. Moreover, as discussed above, heritability studies on ASD prevalence in monozygotic twins taking chorionicity into account could improve the dissection of genetic and intrauterine environmental effects, further increasing our understanding of the role of genetic influence. Higher concordance in monochorionic compared to dichorionic twins would support the role of placental dysfunction in the latter incidence of ASD.

Finally, and perhaps most importantly, CSF IGF-1 level determination could be extended to at-risk neonates 3–6 months old in order to test the hypothesis of early elevated levels of this neurotrophic factor, as such an analysis has not been reported yet. The lower concentration of arginine vasopressin, but not oxytocin, in the CSF of 0–3-month-old neonates in high concordance with idiopathic ASD diagnosis in a quasi-prospective re-evaluated random population sample [200] supports that neonatal liquor composition could show a clear alteration in ASD from typical development at such an early age. If liquor IGF-1 levels are found elevated in affected neonates, pregestational dietary or antidiabetic interventions could be tested prospectively on ASD incidence.

Limitations of the above hypothesis lie in the lack of incorporation of the paracrine effects of IGF-1 or the complex contribution of IGFBPs, and the role of genetic predisposition in affecting the incidence and the phenotype of emerging ASD, factors that could serve as subjects of further research.

## 3. Conclusions

The most consistent maternal and gestational risk factors of idiopathic ASD overlap with two groups of gestational conditions affecting fetal growth through the insulin-IGF-1 axis. Asynchronous termination of these drivers after birth would lead to a biphasic postnatal dysregulation of IGF-1 tone in the infant, with the early insulin resistance-driven overgrowth in the first life year coinciding with the well-documented and specific accelerated development and growth in ASD. This excess IGF-1 spurt might lead to irreversible structural connectivity abnormalities specific to ASD, followed by slowed maturation of neurocircuits, resulting in lasting network dysfunction. If this concept wins confirmation in at-risk infants, in addition to providing new research avenues, it could support effective early prevention initiatives for this burdensome condition.

## Figures and Tables

**Figure 1 ijms-26-04483-f001:**
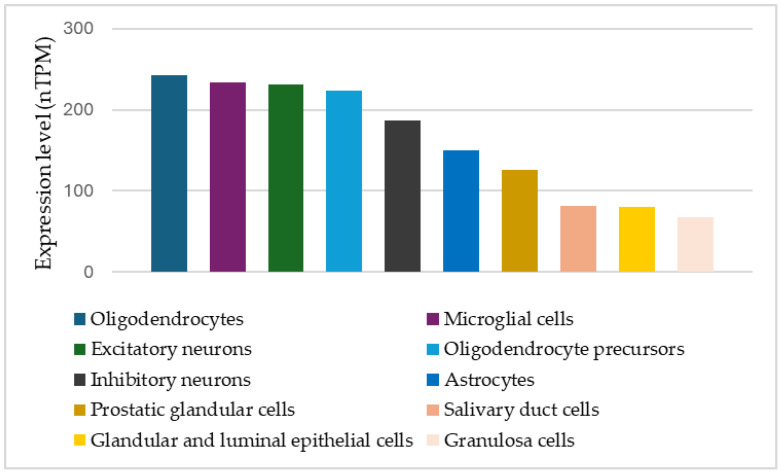
Cell types with highest level of IGF-1 receptor expression. Source: Human Protein Atlas (www.proteinatlas.org accessed on 29 April 2025) [112].

**Figure 2 ijms-26-04483-f002:**
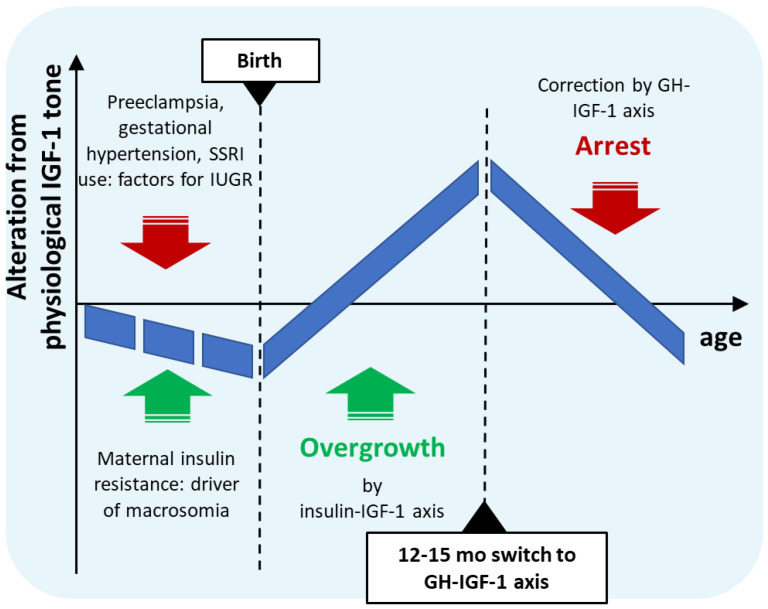
Schematic diagram of age-dependent IGF-1 tone during and following gestational conditions with counteracting effect on fetal growth.

**Table 1 ijms-26-04483-t001:** Proposed phases of perinatal dysregulation of IGF-1 levels in ASD.

Time Period	Hormonal Dysregulation	Effect
Intrauterine development	Insulin resistance and simultaneous growth restriction	Close to normal birth weight
Early postnatal months	Separation from placental unit, release from growth restriction with prevailing insulin-resistance	Restoration of insulin secretion with elevated IGF-1 levels
First year	Insulin-driven IGF-1 overproduction	IGF-1-mediated overgrowth and accelerated neural development
From 12–15 months to ca. 4–5 years	Maturation of GHRH-GH-IGF-1 axis inhibits IGF-1 production	Arrest of growth and brain maturation

## Data Availability

No new data were created or analyzed in this study. Data sharing is not applicable to this article.

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
