# Peer review of "The Possible Role of Postnatal Biphasic Dysregulation of IGF-1 Tone in the Etiology of Idiopathic Autism Spectrum Disorder"

_ijms, 2025, doi:10.3390/ijms26104483_

Round 1

Reviewer 1 Report

Comments and Suggestions for Authors

Summary
The authors propose a potential association between the modulating role of insulin-like growth factor 1 (IGF-1) and autism spectrum disorder (ASD). The topic is timely and relevant, considering the increasing interest in the molecular and therapeutic mechanisms associated with ASD. However, the manuscript requires additional structural and content improvements to enhance its scientific rigor and clarity. Specifically, a Materials and Methods section should be included to detail the literature search strategy and study selection process. Furthermore, a dedicated section on IGF-1 therapy is highly recommended, with a deeper discussion based on the existing literature.

Title
The title should be revised to conform to MDPI’s Instructions for Authors, ensuring that each significant word is capitalized.

Introduction

  • In the Background subsection, the current definition of ASD as a neurodevelopmental disorder should be provided according to the DSM-5. The following reference is suggested: https://doi.org/10.1007/s10803-021-04904-1
  • Line 38: The statement suggesting that no specific genes have been associated with ASD is misleading. Based on multiple studies and resources such as the Malacards database (https://www.malacards.org/), several high-confidence genes are implicated in ASD. These include CHD8, TSC2, NRXN1, MEF2C, CHD2, TRAPPC9, DCC, KAT6A, SHANK3, and SHANK2, among others. I recommend including a short paragraph listing these genes to provide a more balanced view, while also acknowledging the role of environmental influences.
  • The authors should incorporate epidemiological data to highlight the prevalence and growing burden of ASD worldwide.
  • In Subsection 1.2, referencing only the increase of Purkinje cells in the cerebellum is overly reductive. I recommend expanding this section to include broader neuropathological features. The following reference could be particularly helpful: https://doi.org/10.1007/s12015-024-10724-4.

Expression Analysis and Visualization

  • Line 147: Authors are encouraged to include a graphical representation of IGF-1 expression data across relevant tissues or brain regions. This could be done using publicly available databases such as the Human Protein Atlas (https://www.proteinatlas.org/). Presenting the data in the form of a heatmap or barplot would enhance the manuscript’s impact and support the hypothesis more effectively.

Materials and Methods
A dedicated Materials and Methods section should be added to ensure reproducibility. This section should detail:

  • The literature search strategy (databases used, search terms/keywords)
  • Inclusion and exclusion criteria for selected studies
  • Any additional parameters used for data curation or synthesis

IGF-1 Therapy
Given the therapeutic potential of IGF-1, I strongly recommend adding a new section specifically focused on IGF-1–based interventions in ASD. This should include:

  • Mechanisms of action
  • Experimental and clinical outcomes
  • Challenges and future directions

The authors should particularly expand upon the insights presented in Reference 92, which seems central to this theme.

Reviewer 2 Report

Comments and Suggestions for Authors

The manuscript addresses an interesting topic in the field of neuroscience: the postnatal biphasic dysregulation of IGF-1 levels in infants with idiopathic ASD. Although ASD has been studied for years, the theoretical proposal presented by the author in this work provides a new approach to the etiology of this nervous system disorder. The combination of clinical relevance and an established approach makes this work a valuable contribution to its field. Overall, the manuscript is well-structured and written, allowing for guided reading and supported by robust evidence. The conclusions are appropriate for the study's objective and the scientific argument presented. Regarding the cited literature, fundamental studies and reviews recognized in the field are mentioned; a fifth of the references are updated; this could be due to the limited literature on the topic in question. Minor revisions are suggested:

-In lines 264-266, the author is advised to indicate the age range they are referring to when using the terms "older pediatric subjects" and "older children with ASD" to highlight the different age windows and consider the different physiological processes associated with this factor.

-The author is suggested to provide a possible explanation for why there were no differences in CSF levels of IGF-1 between ASD and control subjects; and why are there differences in the reported serum IGF-1 levels in children with ASD? Perhaps the authors of the original reports used different IGF-1 detection tests? Was there a wide age range among the study subjects in the different studies? Etc.

-Indicate at what age maternal age is considered a risk factor for the different clinical manifestations associated with idiopathic ASD.

-Words with Latin roots should be written in italics. Examples: lines 125, 159, and 253, in vivo and in vitro.

-There are several finger errors. Examples: line 199, postanatal; lines 242 and 293, appearence; line 310, awereness.

Reviewer 3 Report

Comments and Suggestions for Authors This manuscript presents a new hypothesis linking biphasic dysregulation of IGF-1 tone to the etiology of idiopathic autism spectrum disorder. The hypothesis is based on existing literatures, integrates several risk factors. However, some areas require revision. 1. The authors should include more experimental evidence such as animal models or discuss potential downstream targets. 2. The central opinion of postnatal biphasic IGF-1 dysregulation relies heavily on indirect evidence. Direct longitudinal data on IGF-1 levels in at-risk neonates are lacking. 3. The hypothesis assumes IGF-1 dysregulation explains all idiopathic ASD cases, but the heterogeneity of ASD may require subtype-specific mechanisms. 4. How do the ASD risk genes interact with IGF-1 pathways? 5. The discussion of IGF-1 as a treatment target is brief, it is better to propose testable interventions. 6. A schematic or table summarizing hormonal regulation shifts and potential disruption points will be benefit. 7. The references are outdated, more recent references should be cited.

Round 2

Reviewer 1 Report

Comments and Suggestions for Authors

Authors addressed all the referee's comments

Reviewer 3 Report

Comments and Suggestions for Authors

No further revision is required.